# Facility Management Costs for Hospital Infrastructures: Insights from the Italian Healthcare System

**DOI:** 10.3390/healthcare13080924

**Published:** 2025-04-17

**Authors:** Michele Dolcini, Sofia Borghi, Marco Rossitti, Andrea Brambilla, Silvia Mangili, Francesca Torrieri, Stefano Capolongo

**Affiliations:** Department Architecture Built Environment Construction Engineering (DABC), Politecnico di Milano, Via G. Ponzio, 31, 20133 Milan, Italy; michele.dolcini@polimi.it (M.D.); sofia.borghi@mail.polimi.it (S.B.); marco.rossitti@polimi.it (M.R.); silvia.mangili@polimi.it (S.M.); francesca.torrieri@polimi.it (F.T.); stefano.capolongo@polimi.it (S.C.)

**Keywords:** healthcare infrastructures, hospital facilities, hospital administration, hospital costs

## Abstract

**Background**: Hospital infrastructures account for a significant portion of healthcare expenditures, yet the factors driving facility management costs (FMCs) remain underexplored, particularly in the Italian context. This study aims to analyze FMCs in hospitals, focusing on utility and maintenance expenses, while providing benchmarking values to support decision making. **Methods**: This study employed a mixed-methods approach, integrating a literature review, financial data analysis, and a case study of 27 hospital facilities in Lombardy. Data on utility and maintenance costs were collected from financial statements and supplemented with targeted questionnaires to enhance precision. Descriptive statistics and parametric cost indicators (e.g., EUR/sqm, EUR/bed) were analyzed to identify trends and disparities. **Results**: FMC increased by an average of 32.90% between 2019 and 2022, with utility expenses constituting 77.45% of total costs and maintenance accounting for 22.45%. Utility costs rose significantly (37.34%), driven by energy and cleaning services, while maintenance costs grew more moderately (18.66%). Cost variability was evident across hospital typologies, with Basic Healthcare Centers averaging 122.86 EUR/sqm compared to 232.66 EUR/sqm for Level II Emergency Hospitals. **Conclusions**: This study highlights significant variability in FMCs across Italian hospitals and underscores the need for benchmarking to optimize resource allocation. Future research should expand the dataset, incorporating extraordinary maintenance costs, and examine management models to enhance cost efficiency. These findings provide actionable insights for policymakers and healthcare administrators to improve hospital infrastructure sustainability and efficiency.

## 1. Introduction

### 1.1. Hospital Facilities Within Healthcare Systems

Hospital facilities represent the core of healthcare organizations, embedding the highest requirements in terms of organizational processes and building factors [1]. As one of the most complex and resource-intensive civil infrastructures, hospitals account for a significant proportion of total healthcare costs at the national level [2]. In the European context, hospital-related expenses constitute the highest share of healthcare expenditures, ranging from 28% in Germany to 48% in Romania, with an average share of 35% [3]. This study focuses on the Italian context, positioned in the highest section with 43.5%. As a pivotal element in the healthcare service, optimizing costs in the sector can positively influence other interconnected components of the system [4].

### 1.2. The Italian Hospital Building Stock

The hospital building stock plays a critical role in cost containment strategies within the healthcare delivery sector, as well as in mitigating the environmental and economic impacts of healthcare activities [5]. Maintaining high standards in hospital infrastructure requires the modernization of the existing building stock, facilitating the adoption of more innovative models better suited to contemporary contexts [6]. This process is essential for optimizing operational costs and improving the overall efficiency and sustainability of healthcare facilities [7].

Due to national reforms and optimization targets, Italian hospital facilities experienced a significant decrease in terms of numbers over the last few decades. From 1954 to 2021, the total number of hospitals decreased by 53.7%, with public hospitals experiencing a more dramatic reduction of 64%. Similarly, in the past fifty years, from 1971 to 2021, the number of healthcare institutions dropped from 2253 to only 1060. As of the most recent census in 2021, the number of hospitals represented 74.4% of the total compared to twenty years earlier (Annuario Statistico del SSN, 1955–2021) (Figure 1). Despite a growing population, the number of hospital facilities has declined, leading to a reduced ratio of hospitals available to meet the needs of the population (Figure 2). Specifically, the total number of available beds, including both public and accredited hospitals, decreased significantly from 295,809 in 2001 to 236,481 in 2021 [8].

A report by the Observatory on Italian Healthcare Companies and Systems, based on 2020 data from the Ministry of Economy and Finance (MEF), provided a comprehensive overview of the aging real estate assets within the National Health Service (SSN), encompassing both hospital and non-hospital infrastructures. As highlighted in the report, most of the building stock owned by Italian healthcare organizations has exceeded its optimal lifespan of 50 years. Over 60% of the building stock in the Italian regions of Piedmont, Lombardy, Liguria, Tuscany, Marche, Lazio, and Sicily, which together account for 52.39% of the population [9], was built before 1969.

### 1.3. Facility Management in Healthcare Infrastructures

Facility management in hospitals plays a fundamental role in ensuring the efficiency, safety, and sustainability of healthcare infrastructures. Facility management has been defined as “an integrated approach to maintaining, improving and adapting the buildings of an organisation in order to create an environment that strongly supports the primary objectives of that organisation” [10]. It presents unique challenges compared to other real estate sectors, such as education, public sector buildings, and office facilities. First, the management of healthcare facilities, given their nature of complex and dynamic environments, requires a robust specialized knowledge base for making informed decisions. Furthermore, facility management for complex assets like hospitals involves several activities, ranging from managing day-to-day operations, preventive, routine, and corrective maintenance, ensuring certification and compliance with international standards, monitoring energy management, space planning, and facility utilization analysis [11]. The complexity stemming from dealing with multiple activities often clashes with the resource constraints, affecting many hospital realities, especially the public ones [12]. These constraints negatively impact the quality and availability of services, as well as the capacity to meet the demands for timely structural and functional upgrades in healthcare facilities, which require significant investments [13].

In this context, the consequent demand for management efficiency and cost containment in healthcare has led hospital administrators to focus on optimizing FM costs through preventive strategies, energy-efficient solutions, and data-driven asset management through the integration of advanced technologies such as Building Information Modeling (BIM), artificial intelligence (AI), and the Internet of Things (IoT) [14,15]. Such a focus calls for devoting particular attention to facility management concerning maintenance, one of the most challenging and costly activities within hospitals [16]. Facility management, indeed, ensures the operational continuity of hospital structures by minimizing disruptions, preserving the quality of patient care, and mitigating the effects of infrastructure obsolescence [17]. Infrastructure obsolescence, a key factor influencing hospital operational efficiency, can be categorized into functional and technological obsolescence [18]. Functional obsolescence occurs when changes in healthcare demands, operational workflows, and regulatory requirements outpace the capabilities of the existing infrastructure, making it inadequate for current needs. Furthermore, technological obsolescence pertains to the outdated nature of building systems and technical components. Hospitals that are not designed to accommodate modern medical technologies or advanced infrastructure solutions may struggle to integrate innovations, ultimately reducing their adaptability and effectiveness [19,20]. Addressing these challenges requires a forward-looking approach to facility management, incorporating strategic planning, lifecycle cost analysis, and proactive maintenance frameworks.

### 1.4. Study Objective

The objective of this research is to investigate the facility management costs (FMCs) associated with hospital infrastructures within the Italian healthcare system, with a specific focus on Lombardy, Italy’s most populous region [21]. In particular, this study aims at observing the variation over time to understand the evolution of expenditures and identifying the key factors that contribute to the differences in costs across various hospital facilities, such as their typology, complexity, age, and maintenance. Furthermore, the contribution aims to provide benchmarking values for parametric cost indicators to compare hospital facilities’ expenditures and offer comparative insights for different hospital typologies. Benchmarking is widely utilized by organizations to drive continuous improvements in the management of building assets. Although benchmarking is recognized as a valuable tool for optimizing operational processes in healthcare facilities, its application in the healthcare infrastructure domain remains very narrow, with limited scientific research available. In Italy, benchmarking tools are widely underutilized, since there is a notable lack of comprehensive data on the operational and maintenance costs of healthcare facilities [22]. The results obtained in the contribution aim to provide information for hospital facility managers and healthcare decision makers to optimize resource allocation and improve cost efficiency and strategic planning.

### 1.5. Research Questions

To explore the facility management costs of hospital facilities, this research addresses the following research questions:What are the facility management cost patterns across hospitals with varying levels of infrastructure and organizational complexity?How do facility management costs evolve over time, and what are the main drivers of variability in operational and management costs across different types of hospital facilities?

### 1.6. Research Context

This research focused on the Italian hospital system, with a particular emphasis on the Lombardy region. Lombardy was chosen as the primary case study as a representative example in the Italian healthcare delivery scenario [23]. This region was deemed appropriate for in-depth analysis due to the following: (i) the significance in the number of hospitals built during different periods; (ii) the variety of hospital types, including new constructions and older facilities, which allows for comparative analysis; (iii) it is the Italian region with the highest population and the most significant contribution of private accredited providers in the healthcare system. Furthermore, the region has a diversified mix of urban and rural healthcare infrastructures, as well as a significant presence of public and accredited private providers, making it a robust case for analysis. While the healthcare systems vary across regions, the diversity and scale of the Lombardy system allow for meaningful insights and serve as a proxy for national-level trends [24].

## 2. Materials and Methods

This paper explores various approaches to address the complexity of managing hospital facility costs. The adopted methodology can be categorized into three distinct phases: (i) theoretical, (ii) practical, and (iii) interactive (Table 1). Each phase is unified by the objective of collecting data and generating insights into the facility management costs of hospital infrastructures. In the theoretical phase, the authors conducted a comprehensive review of both the scientific and gray literature to assess the existing knowledge regarding the costs associated with managing healthcare facilities and to determine the Italian building stock. The practical phase involved analyzing case studies from the Lombardy region to establish parametric cost values. Finally, in the interactive phase, data were gathered through direct engagement with hospital officials. This allowed for a more detailed and coherent analysis of parametric values, particularly for those not readily available from publicly accessible financial documents.

### 2.1. Literature Review: Theoretical Phase

The first phase focused on reviewing the scientific literature on facility management within healthcare facilities to explore existing contributions on the topic of hospital management costs in healthcare environments. The analysis was conducted according to the PRISMA protocol for a systematic literature review in June–July 2024 using the scientific databases Scopus, Web of Science, and PubMed (Figure 3). The following keywords were employed: “maintenance cost”, “utilities cost”, “operating cost”, “management cost”, and “facility management cost”, combined with terms defining the healthcare setting, such as “hospital”, “hospital building”, “hospital center”, “healthcare facilities”, and “healthcare infrastructure”. To ensure the relevance of the researched contributions and of the data retrieved, the following exclusion criteria were applied during the literature review: studies prior to the year 2002, with the adoption of the EU common currency. Papers in the field of medicine, clinical practice, and nursing were excluded since these disciplines do not address the research focus on facility management and operational costs. Non-peer-reviewed documents, such as opinion pieces, letters, or editorials, were excluded from the search on scientific databases. The literature review returned a result of (n = 87) papers by reading the title, then 53 were excluded by reading the abstract and keywords, and a final result of (n = 34) papers was included in the final full text, in addition to (n = 3) contributions from the gray literature. After full-text screening, only the results of (n = 7) papers were considered. The second step involved the analysis of the available gray literature by searching online for the keywords “facility management cost”, “operational costs”, “O&M costs”, and “Italy”.

### 2.2. Data Collection: Practical and Interactive Phase

This phase focused on gathering hospital operational cost data from balance sheets from a selection of healthcare organizations within the Lombardy region. A total of 27 hospital organizations (“Aziende Socio Sanitarie Territoriali” ASST) were analyzed. Data on hospital infrastructure management costs were gathered by collecting costs and operational data from hospital balance sheets and financial statements, available on the official websites, covering the years 2018 to 2022. The timeframe was chosen to analyze the most recent data available and to mitigate fluctuations in expenses caused by the extraordinary circumstances that affected Italy during these years. The following balance sheet items were included in the analysis: 1. Utilities: non-healthcare service purchases; cleaning; heating; waste disposal; electricity; water, gas, and fuel; other utilities. 2. Outsourced ordinary maintenance and repair: routine maintenance and repair of buildings and their related assets; routine maintenance and repair of equipment and machinery; routine maintenance and repair of furniture and vehicles; other maintenance and repair activities (Table 2).

### 2.3. Interactive Phase: Case Studies Analysis

The interactive phase of the analysis, through its focus on specific case studies, aimed to collect detailed data directly from hospital facilities rather than relying solely on aggregated values obtained from the financial statements of healthcare organizations. In balance sheets and financial statements, available online for all healthcare public companies, facility management costs (FMCs) were not exclusively related to hospital facilities, but they also included buildings used for other purposes. To isolate the specific costs related to hospital facilities, a customized survey-based data collection form (Google Forms) was developed. A customized data collection form was designed and distributed to all technical departments in selected public hospitals across Lombardy (n = 132). For the description analysis, key indicators such as mean, median, standard deviation, and minimum and maximum values were calculated to summarize the distribution of facility management costs, both in terms of EUR/m^2^ and EUR/bed, across hospital types and years. For group comparison analysis, to compare cost values across hospital typologies (Basic, DEA I, and DEA II), the authors used ANOVA for normally distributed data and the Kruskal–Wallis test for non-normally distributed variables. Finally, to determine the relationship between facility characteristics (e.g., age, surface area, and number of beds) and cost, variables were examined using Pearson or Spearman correlation coefficients, depending on the distributional characteristics of the data.

#### 2.3.1. Sample Description

A customized data collection form was designed and distributed to all technical departments in the selected public hospitals across Lombardy (n = 132). Data were collected from (n = 27) hospitals, corresponding to a response rate of 20.4%. The facilities vary in complexity and typology, including Basic Healthcare Centers, Level I Emergency Hospitals (DEA I), and Level II Emergency Hospitals (DEA II). These hospitals also differ in size (ranging from 65 to 1200 beds), gross floor area (from 6013 m^2^ to nearly 200,000 m^2^), and building age (constructed between 1907 and 2012). All the facilities are publicly owned and operated by the ASST (Appendix A). This diversity enabled a meaningful comparative analysis across various levels of infrastructural and organizational complexity. The localization of the 27 hospitals included in the case studies’ analysis can be observed in Figure 4.

#### 2.3.2. Sampling Method

The sampling method adopted in this study was a comprehensive outreach to all publicly owned hospital facilities in the Lombardy region. A standardized data collection form was distributed via email and telephone contact to the entire population of 132 public hospitals, with the aim of maximizing participation and ensuring broad representativeness across different facility typologies. 

#### 2.3.3. Questionnaire Design

To ensure the reliability and clarity of the data collection instrument, the questionnaire underwent a two-step validation process. First, the content was developed based on the established classifications used in Italian public healthcare financial reporting, specifically aligning with cost categories outlined in regional and national health accounting frameworks. This provided a consistent basis for capturing facility-related operational costs (e.g., utilities, cleaning, and maintenance) across multiple institutions. Second, the questionnaire was pre-tested with a technical officer from a hospital not included in the final sample. The pilot test was used to assess the clarity of the questions, the appropriateness of the terminology, and the feasibility of retrieving the requested data from internal systems. Finally, the questionnaire was structured into the following three sections: 1. Research objectives and context. 2. Facility identification and structural characteristics (e.g., year of construction, gross floor area, and number of beds). 3. Operational cost data from 2018 to 2022. The survey items were derived from national health budget classification systems and adapted to isolate hospital-specific costs. The form was developed in consultation with technical departments and pre-tested with three pilot facilities to ensure clarity, data consistency, and content validity (Table 3).

## 3. Results

### 3.1. Literature Review

As mentioned in the Methodology Section, the search conducted in the scientific databases yielded only limited significant results. The reviewed studies provided partial insights into the research questions. Among the most significant contributions in this area is the work by Sliteen et al., who analyzed the operational costs of 19 French hospitals [22]. Their findings revealed that, on average, 38% of the total operational and maintenance costs in hospitals are attributed to utilities, 29% to maintenance expenses, and the remaining 33% to staffing costs for real estate functions. This analysis underscores the fragmented nature of existing research on hospital operational costs and highlights the importance of conducting further empirical studies to develop comprehensive benchmarks and better understand cost drivers in hospital facility management. The literature review revealed a significant gap in studies focusing on the identification of benchmark values for hospital management costs and the key factors contributing to increases in operational expenses. This lack of research has hindered the development of standardized benchmarks for hospital management costs. Consequently, an inductive approach, involving the empirical observation of hospital facilities, was necessary to address these gaps. Several studies have confirmed the use of gross floor area (GFA) as the primary functional unit for determining management and maintenance costs [25]. Gómez-Chaparro and colleagues established a relationship between maintenance and hospital characteristics, which is that hospitals with more than 25,000 annual stays exhibit lower maintenance costs per area, fewer maintenance reports per bed, and fewer reports per unit area compared to hospitals with fewer stays, while large hospitals show later maintenance costs per unit area and fewer maintenance costs per bed then smaller hospitals, confirming an expected economy of scale principle. Their analysis also addressed the most critical maintenance variables identified: heating, ventilation, and air conditioning (HVAC) systems, refurbishment works, and plumbing installations. Finally, Hassanain and colleagues identified, through a questionnaire with 40 hospitals, the key factors affecting rising maintenance costs, and the most critical factor affecting costs was linked to some problems in the design and construction phase, followed by the absence of quality control during system installations [26].

In the Italian context, there is a noticeable absence of scientific contributions specifically addressing management and maintenance costs. However, the review of the gray literature described in the Methodology Section provided some results, and three technical reports were analyzed: the first published by the AGENAS (National Agency for Regional Healthcare Services), the IRES (Institute for Economic and Social Research of the Piedmont Region), and the OASI 2021 Report. The OASI Report highlights a linear relationship between the costs incurred for the routine maintenance of buildings, systems, and machinery, and the aging of the building stock [27]. As mentioned earlier in this section, greater wear and tear on an asset’s conventional useful life should correspond to a higher intensity of maintenance requirements. This underscores the importance of integrating lifecycle considerations into facility management strategies to address the aging infrastructure and optimize resource allocation. The IRES contribution from a sample of Italian hospitals provided a value for maintenance cost per square meter parametric value based on 2016 data. The study estimates an average maintenance cost of EUR 58.3 per square meter per year. These costs are influenced by the obsolescence level of hospital facilities. The study provides differentiated ranges of annual costs based on obsolescence: between 30 EUR/sqm per year for new infrastructures and up to 100 EUR/sqm for the hospitals with the highest level of obsolescence [18].

An AGENAS analysis similarly analyzed a sample of 29 hospital facilities across five Italian regions and provided an average operating cost of 45.05 EUR/sqm. Maintenance accounted for the largest share of operational costs at 47%, followed by electricity (30%), heating (19%), and water (4%). The study also highlighted regional differences between urban areas and suburban or rural areas (27.4%). Building typologies and characteristics also influence operational expenses, with hospitals with pavilion-style layouts reporting higher operational costs (48.85 EUR/sqm) compared to monoblock structures (40.63 EUR/sqm). Additionally, older buildings showed higher operational costs (52.79 EUR/sqm) than newer facilities (37.29 EUR/sqm) with the analyzed sample. These findings emphasize the impact of design factors on hospital operating costs [28].

### 3.2. An Aging Italian Hospital Building Stock

The state-of-the-art assessment of a sample of 500 facilities in Italian healthcare, conducted through the information available online, revealed a significant proportion of healthcare institutions exceeding 50 years of age, the typical useful life limit [29]. Notably, 29% of these structures were built between 1941 and 1970, making this period the most significant in terms of hospital development. Buildings constructed between 1901–1940 and 1971–2000 each account for 22%, while only 19% of hospitals were built after 2001, reflecting a decline in recent construction efforts. Meanwhile, 8% of hospitals date back to before 1900, showcasing a portion of the building stock that is over a century old. This distribution highlights potential challenges related to aging infrastructure, as nearly 60% of these facilities were built before modern construction standards, possibly requiring significant upgrades to meet current safety, functionality, and sustainability requirements.

The findings highlight a significant issue: many hospitals across Italy are housed in outdated and aging structures, making it increasingly challenging for them to adapt to the rapid advancements in modern medicine. These facilities, often inherited from decades of healthcare history, face significant maintenance and operational challenges that lead to substantial and frequently unsustainable costs. The situation not only affects the quality of care provided but also places a considerable strain on the financial resources of healthcare facilities, as significant funds must be allocated to maintain buildings that are often inadequate for current needs. Consequently, there is an urgent need to renew and modernize the national hospital infrastructure.

The data presented in Figure 5, concerning the construction year of Italian hospital buildings, refer to a broader sample of 500 facilities for which publicly available information was collected. This investigation was exclusively aimed at identifying the year or period of construction, without considering building typology or specific hospital characteristics, and relied solely on data accessible through institutional websites. These data are entirely distinct from the primary cost analysis sample, which includes 27 hospitals selected through a targeted, questionnaire-based investigation.

### 3.3. Hospital Organizations’ Facility Management Costs

An analysis of the Lombardy hospital facilities examined reveals that 70% have exceeded the useful life of 50 years, with construction dates ranging from 1907 to 2012. The hospitals studied have an average gross floor area of 56,193 sqm and a capacity ranging from 65 to 1200 beds. Despite the considerable diversity of the sample, it was still possible to identify the average utility and routine maintenance costs (Figure 6).

The analysis illustrates the aggregated trend in facility management costs across the 27 hospital facilities included in the analysis. The *y*-axis represents the total annual costs expressed in millions of euros (€M). This aggregation provides a macro-level perspective on how overall facility management expenditures have evolved from 2019 to 2022 across the entire sample. The graph clearly demonstrates a consistent increase in costs over the observed period, with a notable surge between 2021 and 2022. This trend is largely attributed to rising energy prices and pandemic-related impacts on operational services. By presenting total costs rather than average or per-unit values, this figure supports a broader understanding of the cumulative financial burden borne by hospital infrastructures in the region.

### 3.4. Interactive Phase: A Questionnaire on Operational Costs of Healthcare Infrastructures

Following the analysis of the financial statements of hospital organizations, a targeted questionnaire was implemented to enhance the precision of the analysis by selecting a specific sample of hospital facilities. The objective was to gather more detailed and granular information compared to what was available from the financial statements. While financial statements provide an aggregated and retrospective view of data, enabling large-scale application at the national level, they are inadequate for in-depth scientific investigations into facility management (FM) parameters at the level of individual hospital facilities. Through the questionnaire and the data collected, a range of parameters was derived for analysis. These included costs per square meter for aggregated items (e.g., total utility expenses), total routine maintenance costs, and the overall running costs for infrastructure management. Firstly, the data collected reveal that utility expenses constitute the primary cost component, accounting for 77.45% of the total management costs. In contrast, routine maintenance expenses represent a smaller proportion, amounting to 22.45% of the total. These findings underscore the significance of utilities in hospital infrastructure management and provide valuable insights for optimizing cost efficiency across facilities.

An analysis of cost data for hospital facilities over five years (2018–2022) reveals a consistent upward trend in expenditures. During this timeframe, utility costs increased by 10.64%, with a notable decline in 2019 attributed to a 46.92% reduction in heating expenses. Conversely, a substantial 34.19% rise in heating-related costs drove the overall increase in 2022. Other utility expenses displayed steady growth, albeit at varying rates. Ordinary maintenance costs exhibited consistent growth until 2021, followed by a slight decrease between 2021 and 2022. A significant surge in maintenance expenses occurred between 2020 and 2021, primarily due to the adaptation of buildings and systems to address the challenges posed by the COVID-19 pandemic. This analysis underscores that utility costs are the most significant and impactful component of hospital infrastructure management services, while ordinary maintenance costs showed less dramatic fluctuations within the broader trend. From 2018 to 2022, the total management services costs rose by 16.48%, with a cumulative increase of 32.90% observed since 2019. Notably, this period includes the economic effects of the COVID-19 pandemic, which disrupted supply chains, increased demand for cleaning and sanitization, and led to inflationary pressures in the energy and service sectors. Additionally, policy-driven adjustments, including stricter hygiene regulations and energy consumption standards, may have further elevated costs, especially in hospital facilities operating with outdated systems. As such, part of the cost variability observed in this study can be reasonably attributed to time-bound conditions (Figure 7).

Figure 8 presents the average parametric costs for hospital facility management, disaggregated into three main components: utilities costs, ordinary maintenance, and overall management services. The left panel displays the cost distribution expressed in euros per square meter (EUR/m^2^), while the right panel shows the same categories expressed in euros per hospital bed (EUR/PL). The average total costs of utilities and maintenance amount to EUR 161.58 per sqm, ranging from EUR 66.67 to 341.44 per sqm. These expenses consist of utility services, with an average value of 128.78 EUR/sqm, and maintenance costs, averaging 32.8 EUR/sqm on an annual basis.

### 3.5. Variability and Trends in Hospital Facility Management Parametric Costs

The descriptive analysis revealed significant variability in parametric management costs across the analyzed hospital facilities, influenced by the type and complexity level of the structures. The classification of different hospital typologies are defined as Basic Healthcare Centers, Level I Emergency Departments (DEA I), and Level II Emergency Departments (DEA II) and are regulated by Ministerial Decree 70/2015. Basic Healthcare Centers (“Presidi di base”) reported an average parametric cost of 122.86 EUR/sqm. These facilities generally have simpler layouts, lower patient throughput, and reduced technical infrastructure, which translate into lower energy consumption, minimal specialized cleaning requirements, and simplified maintenance routines. In contrast, hospitals classified as Level I Emergency Departments (DEA I) and Level II Emergency Departments (DEA II) demonstrate higher average values, amounting to 159.10 EUR/sqm and 232.70 EUR/sqm, respectively. These differences reflect the greater infrastructural and operational complexity required by higher-level facilities (Table 4).

When considering parametric cost per bed, the data highlights significant differences in facility management costs across healthcare facilities of varying complexity. The average costs for Basic Healthcare Centers are the lowest, with a parametric cost of EUR 122.86 per square meter and an average cost per bed of EUR 18,509.24. These values reflect the relatively simple infrastructure and operational requirements of these facilities (Table 5).

Further statistical analysis provided a more detailed overview of the data distribution. The overall mean parametric cost was 151.47 EUR/sqm, with a median value of 137.31 EUR/sqm, suggesting a moderately right-skewed distribution. The minimum observed value was 66.67 EUR/sqm, while the maximum reached 340.92 EUR/sqm, resulting in a total range of 274.25 EUR/sqm. The standard deviation, calculated as 79.45 EUR/sqm, highlights significant variability in costs, indicative of structural and operational differences among the analyzed facilities (Table 6).

Considering the total expenses incurred for facility management, an increase of 32.90% was observed over the period of 2019–2022. In this range, all 27 hospital facilities involved in the analysis provided data. Among the facilities analyzed, only one recorded stable or negative costs (−1.9%), while the highest increase reached 119%. The analysis particularly highlighted a more significant rise in utility-related expenses, with a total increase of 37.34%, primarily driven by energy costs and cleaning services. This trend reflects the impact of years marked by the COVID-19 pandemic. In contrast, maintenance expenses exhibited a more moderate yet still notable increase of 18.66% over the same period. These findings underscore the varying drivers of cost growth in hospital facility management and provide valuable insights for targeting resource optimization. This trend reflects system-wide dynamics in the healthcare sector, influenced by factors such as rising energy costs, the need for extraordinary maintenance, and the adaptation of facilities to new operational requirements, including those arising from the COVID-19 pandemic. Larger and more complex hospital facilities experienced significant absolute cost increases. However, smaller facilities also showed substantial percentage increases, indicating that operational management costs are systematically growing across all facilities in the sample.

Another key variable analyzed is the conventional age of hospitals. While a general inverse relationship is observable—whereby newer facilities tend to report lower operational costs—this trend shows considerable variability. Such dispersion suggests that building age alone does not fully explain cost differences and that other structural and managerial factors may play a critical role. For instance, hospitals with similar years of construction display substantial cost differences, which may be influenced by divergent building typologies (e.g., monoblock vs. pavilion layout), variations in surface area per bed, energy efficiency measures, and the degree of outsourcing of facility services. Additionally, some older hospitals may have undergone recent minor retrofits or infrastructure upgrades, which mitigate their operational inefficiencies, while others continue to function with legacy systems that drive up maintenance and energy expenses. These findings reinforce the need to adopt a multivariate analytical approach in future studies when assessing cost determinants (Figure 9).

## 4. Discussion

This contribution addressed the management costs of Italian healthcare facilities with different approaches. The findings from this research provide critical insights into the operational costs associated with hospital infrastructure management in the selected sample of 27 hospitals over the period of 2018–2022. The analysis highlights significant variability in costs across facilities and over time, reflecting the diversity of the Italian healthcare building stocks in terms of operational needs, infrastructure age, technology equipment, and the level of complexity. The emerging disparities in cost efficiency among hospitals also suggest opportunities for benchmarking and the adoption of best practices.

The comparison between the data collected from the empirical survey and the findings from the literature review suggests that hospital management costs in Italian facilities tend to be higher. Specifically, the average operational costs observed in the Italian sample exceed those reported in the international studies analyzed in the literature. However, this disparity requires further investigation, considering the need for comparable data in terms of the time of data collection and the complexity of hospital typologies. A more refined analysis accounting for these variables would provide a clearer understanding of the factors influencing cost differentials and provide a robust benchmarking framework.

The findings shed light on cost variability across hospital typologies, operational dynamics, and structural factors, answering the following two primary research questions: (i) What are the costs associated with the facility management of hospitals of different complexities? (ii) How do facility management costs evolve over time, and what are the main drivers of cost variability across different types of hospital facilities? As presented in the Results Section, facilities classified as Basic Healthcare Centers report the lowest average costs compared to Level I and Level II hospitals, showing progressively higher costs associated with their activities and greater operational demands. These results align with previous findings [22,26], which emphasized the correlation between hospital complexity and FMC due to advanced infrastructure and service demands. The data collected in this study differ from the literature regarding the proportion of maintenance costs, as the studies reviewed include extraordinary maintenance costs within their calculations.

While the results provide valuable parametric benchmarks for FM costs in hospital infrastructure, it is crucial to acknowledge that the findings are based on a limited sample of 27 hospitals in the Lombardy region, corresponding to a 20.4% response rate. Although this sample includes a heterogeneous mix of hospital typologies and sizes, the relatively low participation constrains the generalizability of the parametric values (EUR/m^2^ and EUR/bed) both within the region and nationally. Therefore, the results and the values presented should be interpreted as exploratory and indicative rather than statistically representative of the broader healthcare system.

## 5. Conclusions

In this study, through an empirical analysis of a sample of Italian hospitals, the authors explored the feasibility of developing benchmarking metrics to evaluate the facility management costs of hospital assets. The investigation conducted for this contribution on hospital facility management costs has confirmed the fragmented and uneven nature of the Italian hospital building stock, highlighting the significant infrastructural and managerial differences across these facilities.

This study acknowledges some limitations. One of the primary limitations lies in the use of data extracted from hospital financial statements. As discussed in the text, this level of analysis enables a broader national sampling due to the public availability of such data and facilitates the identification of trends. However, it presents constraints since the financial statements of hospital organizations aggregate facility management costs for both hospital and non-hospital healthcare facilities. A significant limitation of this study is thus the limited number of case studies analyzed. Additionally, the scope of the research was geographically limited to the Lombardy region, which, despite being the most populous Italian region and possessing the most extensive hospital numbers, limits the generalizability of the findings to a national context.

Moreover, this study did not delve deeply into hospital management models or the relationship between in-house management and the outsourcing of certain activities. In light of these considerations, future research can be developed along the following three directions: (i) Introducing longitudinal studies that extend beyond the Lombardy region and incorporate cross-country comparisons to enhance the generalizability and contextual relevance of the findings. (ii) Including costs related to extraordinary maintenance to provide a more comprehensive overview. (iii) Investigating hospital management models, particularly examining the differences between internal management and outsourcing for activities such as maintenance, cleaning, energy production, and the management of climate control and lighting systems. This would help assess potential discrepancies in cost and performance between internally managed operations and those entrusted to external facility managers.

This work represents an initial and innovative attempt to map parametric cost indicators based on real operational data, collected directly from healthcare organizations. Given the scarcity of prior benchmarks in this domain, the research prioritizes the establishment of empirical reference values and trends that may serve as a foundation for further investigations. Therefore, the focus has been placed on descriptive and exploratory analysis, which allows for the identification of cost patterns and key areas of variability across hospital facilities of different types and complexity levels. By providing valuable benchmarks and actionable insights for hospital facility managers, our contribution offers preliminary guidance on the optimization of facility management, the allocation of public resources, and the enhancement of the efficiency of hospital organizations. Future research may build upon this work by expanding the sample, applying inferential methods, and integrating multivariate models to explore causal relationships.

## Figures and Tables

**Figure 1 healthcare-13-00924-f001:**
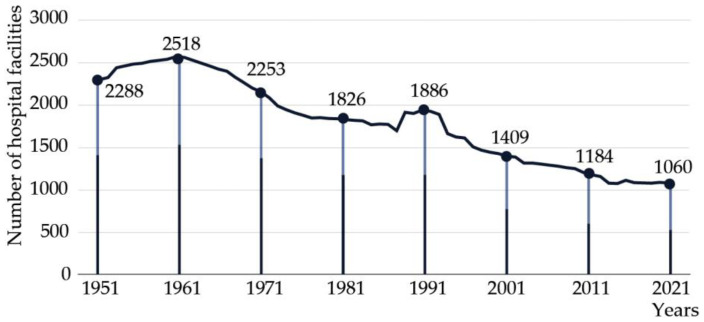
Decline in the number of public and private hospitals (Annuario Statistico del SSN).

**Figure 2 healthcare-13-00924-f002:**
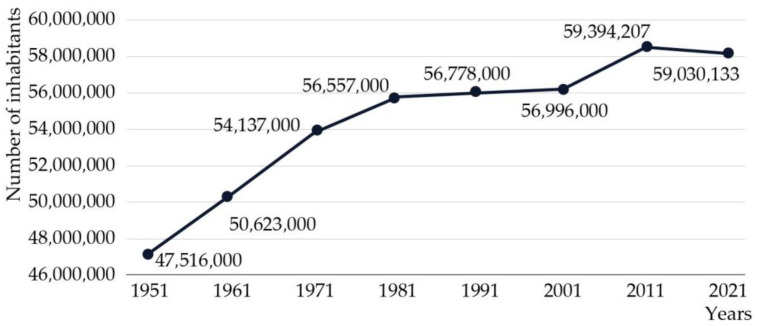
Italian population growth (ISTAT, 1951–2021).

**Figure 3 healthcare-13-00924-f003:**
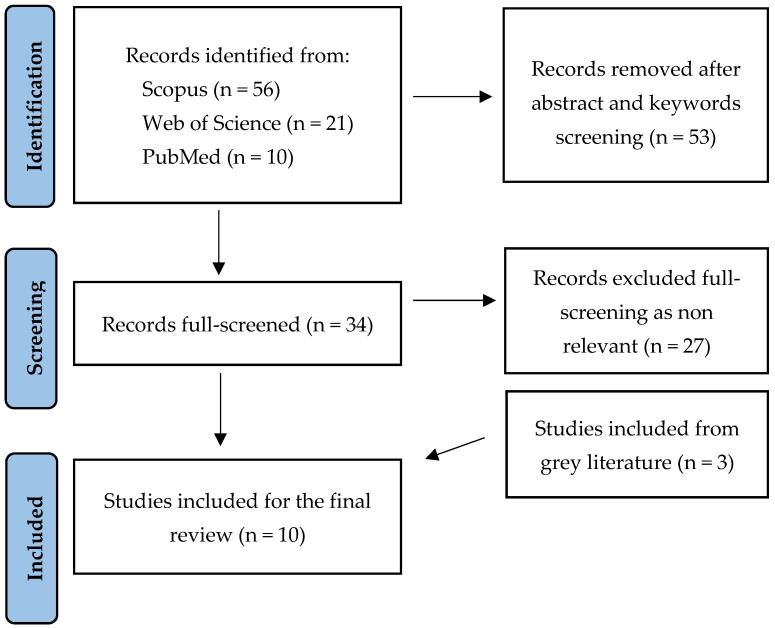
PRISMA chart for literature review.

**Figure 4 healthcare-13-00924-f004:**
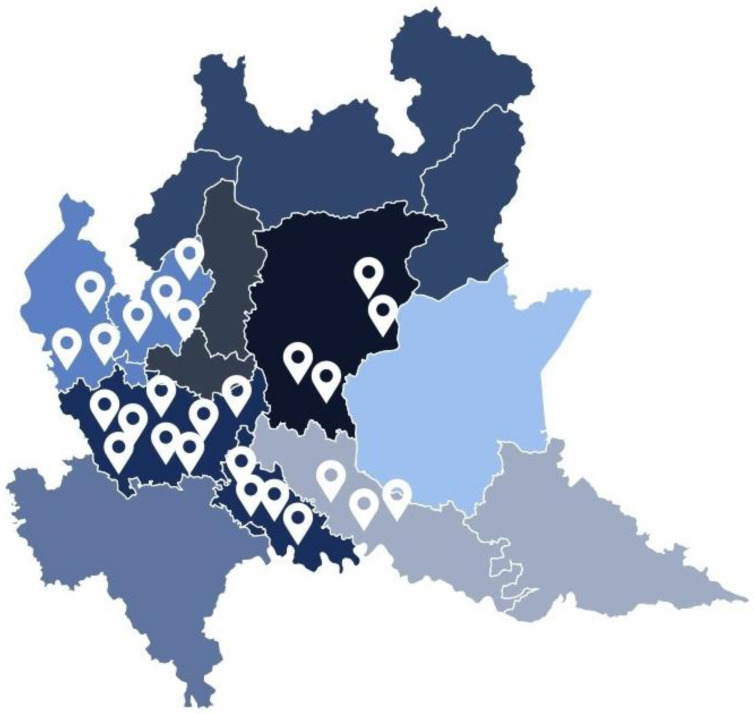
Geographic distribution of sampled hospitals across Lombardy’s health districts.

**Figure 5 healthcare-13-00924-f005:**
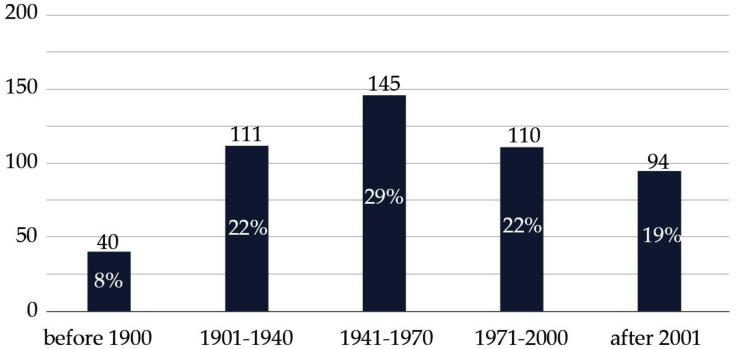
Year of construction of Italian hospitals, based on a sample of 500 hospitals collected from healthcare companies’ websites.

**Figure 6 healthcare-13-00924-f006:**
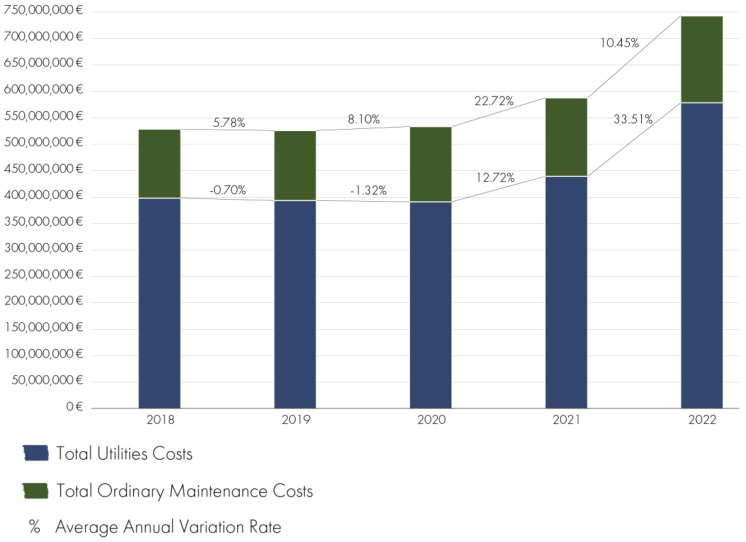
Trend of FM costs across the analyzed hospital sample (2018–2022, EUR).

**Figure 7 healthcare-13-00924-f007:**
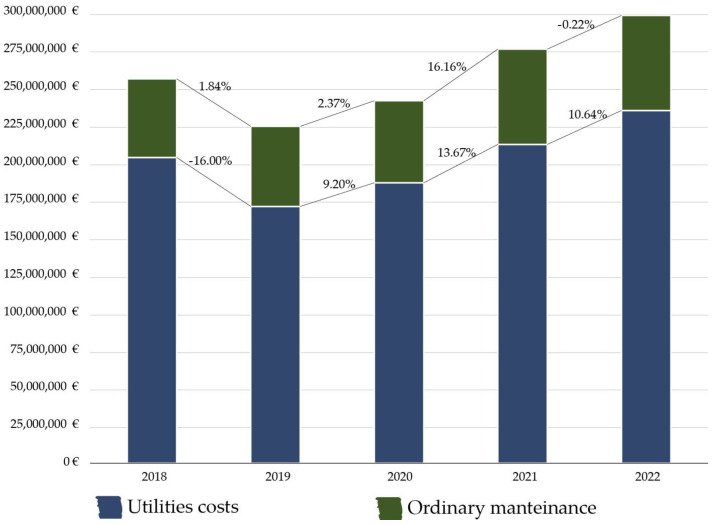
Annual trend of utilities and ordinary maintenance costs (2018–2022, EUR).

**Figure 8 healthcare-13-00924-f008:**
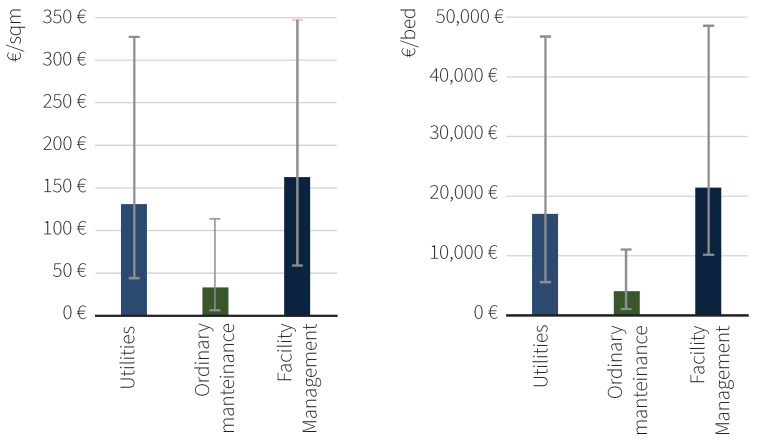
Average facility management costs by type and complexity of hospital (EUR/m^2^) and EUR/bed.

**Figure 9 healthcare-13-00924-f009:**
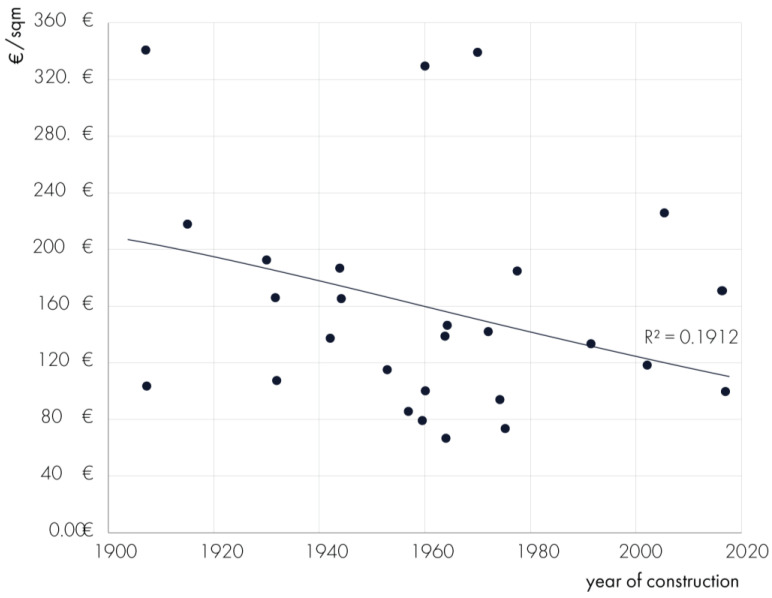
The relationship between operational costs per sqm and the year of construction of healthcare infrastructures.

**Table 1 healthcare-13-00924-t001:** Research methodological approach in three phases.

Approach	Methods	Source	Data Source
1. Theoretical	Literature Review	Scopus, WoS, PubMed, and Gray Literature	Secondary
2. Practical	Healthcare Organization	Balance Sheet	Secondary
3. Interactive	Case Studies	Questionnaire	Primary

**Table 2 healthcare-13-00924-t002:** Key inputs collected for operational and maintenance cost assessment.

Input	Definition
Total utilities cost	The total annual cost of energy consumed by the hospital site, inclusive of the cost items: heating, waste disposal, cleaning, electricity utilities, water, gas, fuel, and other utilities.
Total maintenance cost	Total costs for external services for the provision of maintenance services, including cost of routine maintenance and repair of the building, cost of routine maintenance and repair of systems and machinery, cost of routine maintenance for furniture and equipment, and other maintenance and repair costs.
Total operational and maintenance costs	Total utilities costs + total maintenance costs

**Table 3 healthcare-13-00924-t003:** Variables collected through the survey.

Variable	Definition
Year of Construction	The year in which the hospital was built or significantly renovated. For hospitals with multiple buildings, the average year of construction, weighted by the gross floor area, was considered.
Gross Floor Area	The area within the perimeter of the outside walls of a building as measured from the inside surface of the exterior walls, with no deduction for hallways, stairs, closets, thickness of walls, columns, or other interior features; this is used in determining the required number of exits or in determining occupancy classification.
Number of Beds	The total number of available hospital beds, which is used to calculate the operational costs per bed.
Number of Floors	The total number of levels in the hospital building was considered.
Building Typology	Building classification based on their architectural design, layout, and functional characteristics.

**Table 4 healthcare-13-00924-t004:** Facility management costs (€/sqm) by hospital typology.

Hospital Typology and Complexity	EUR/sqm
Basic Healthcare Centers	EUR 122.86
Level I Emergency Hospital (“DEA di I Livello”)	EUR 159.10
Level II Emergency Hospital (“DEA di II Livello”)	EUR 233.70

**Table 5 healthcare-13-00924-t005:** Facility management costs (€/bed) by hospital typology.

Hospital Typology and Complexity	EUR/Bed
Basic Healthcare Centers	EUR 18,509.24
Level I Emergency Hospital (“DEA di I Livello”)	EUR 24,272.40
Level II Emergency Hospital (“DEA di II Livello”)	EUR 35,458.13

**Table 6 healthcare-13-00924-t006:** Descriptive statistics of hospital total utilities and maintenance costs.

Mean	EUR 151.47
Median	EUR 137.31
Standard Deviation	EUR 79.45
Minimum Value	EUR 66.67
Maximum Value	EUR 340.92

## Data Availability

The data presented in this study are available on request from the corresponding author due to the large amount of data and privacy concerns.

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
