# Peer review of "Facility Management Costs for Hospital Infrastructures: Insights from the Italian Healthcare System"

_healthcare, 2025, doi:10.3390/healthcare13080924_

Round 1
Reviewer 1 Report
Comments and Suggestions for Authors
Minor Comment
-
- Consider breaking down the Background, Methods, Results, and Conclusions more explicitly for clarity.
- The sentence "These findings provide actionable insights for policymakers and healthcare administrators..." is strong but vague—what specific actions do you recommend?
-
Figures & Tables:
- Figure 4 (hospital locations) could be improved by adding more details (e.g., color coding by facility type).
- Table 4 & 5: It would be helpful to indicate the standard deviation for cost values to show variation across hospital types.
Conclusion:
- The discussion on limitations is strong but could be expanded to suggest specific improvements in data collection methods for future research.
- The future research directions should emphasize longitudinal studies beyond Lombardy and cross-country comparisons.
-
Methodology & Data Justification:
- While the methodology is robust, further details are needed regarding the selection criteria for the 27 hospitals in Lombardy. How were they chosen? Were there any biases in selection?
- The questionnaire used in the interactive phase is mentioned, but there is little detail on the questions included. Adding sample questions or a summary of the survey instrument would enhance transparency.
- The time period chosen (2018–2022) covers fluctuations due to COVID-19. Were any normalization techniques used to account for this anomaly?
-
Benchmarking & Comparisons:
- The study provides useful parametric cost indicators (€ per sqm, € per bed), but lacks international comparisons. How do these values compare to those in other European healthcare systems?
- The comparison across hospital typologies (Basic, Level I, Level II Emergency) is insightful, but what specific factors drive the cost differences beyond infrastructure complexity?
-
Results Interpretation & Implications:
- The results indicate a 32.90% increase in costs between 2019 and 2022, mainly due to rising energy and cleaning costs. Could policy changes or inflationary trends have contributed to this? Additional discussion on macroeconomic factors would strengthen the interpretation.
- The impact of hospital building age on facility management costs is significant, but were any renovation efforts considered in the analysis? Some older hospitals may have undergone modernization, influencing their costs.
- References:
- Please add the references in covid 19 sentence in page number 12
- 1.Shakeel S, Rehman H, Hassali MA, Hashmi F. Knowledge, attitude and precautionary practices towards COVID-19 among healthcare professionals in Karachi, Pakistan. The Journal of Infection in Developing Countries. 2020 Oct 31;14(10):1117-24.
- 2. Rehman H, Aziz S, Amjad S, Zehra A, Rafique R, Masood P. Impact of rumors, conspiracy and intuition about covid-19 vaccine in Karachi, Pakistan. RADS Journal of Pharmacy and Allied Health Sciences. 2023 Jul 31;1(2):86-93.
Author Response
Summary |
|
|
Thank you very much for the review of this manuscript. The comments have been very valuable to improve the manuscript. On behalf of all authors, please find the detailed responses below and the corresponding revisions/corrections highlighted/in track changes in the re-submitted files.
|
||
Point-by-point response to Comments and Suggestions for Authors |
||
Comments 1: Methodology · While the methodology is robust, further details are needed regarding the selection criteria for the 27 hospitals in Lombardy. How were they chosen? Were there any biases in selection?
|
||
Response 1: Methodology - Thank you for pointing this out. The 27 hospitals in Lombardy were those responding to the survey, which was sent to all public hospitals in the region (n=132), corresponding to a response rate of 20.4%. The questionnaire structure has been updated. No normalization technique has been adopted for adjusting to fluctuations.
|
||
Comments 2: Benchmarking & Comparisons |
||
· The study provides useful parametric cost indicators (€ per sqm, € per bed), but lacks international comparisons. How do these values compare to those in other European healthcare systems? · The comparison across hospital typologies (Basic, Level I, Level II Emergency) is insightful, but what specific factors drive the cost differences beyond infrastructure complexity? Response 2: Benchmarking & Comparisons We acknowledge the reviewer’s observation regarding the absence of direct international comparisons. While our primary focus was to develop nationally contextualized benchmarks for facility management costs within the Italian healthcare system, we recognize the value of placing these findings within a broader European context. However, the literature review did include a few notable international studies and provided the results, while no specific comparable parametric values are available (Par. 3.1). For instance, Sliteen et al. (2011) analyzed operational and maintenance costs in hospitals and reported that utilities accounted for approximately 38% of O&M costs, maintenance for 29%, and staffing for 33%—a pattern that aligns in part with our findings, where utilities dominate FMC at 77.45%. Similarly, Hassanain et al. (2013) provided parametric estimates for Saudi hospitals, while Gómez-Chaparro et al. examined cost differences in Spanish facilities, supporting the idea that larger hospitals benefit from economies of scale. Comment 3: Results Interpretation & Implications: · The results indicate a 32.90% increase in costs between 2019 and 2022, mainly due to rising energy and cleaning costs. Could policy changes or inflationary trends have contributed to this? Additional discussion on macroeconomic factors would strengthen the interpretation. · The impact of hospital building age on facility management costs is significant, but were any renovation efforts considered in the analysis? Some older hospitals may have undergone modernization, influencing their costs.
Response 3: Results Interpretations & Implications Paragraph 3.4 has been updated according to the review suggestions. Comment 4: Figures & Tables: Figure 4 (hospital locations) could be improved by adding more details (e.g., color coding by facility type). Table 4 & 5: It would be helpful to indicate the standard deviation for cost values to show variation across hospital types.
Response 4: Figures & Tables standard deviation has been provided in the text describing Table 4, 5 and 6. |
Comment 5: Reference
Response 5: The reference has been added
- Shakeel S, Rehman H, Hassali MA, Hashmi F. Knowledge, attitude and precautionary practices towards COVID-19 among healthcare professionals in Karachi, Pakistan. The Journal of Infection in Developing Countries. 2020 Oct 31;14(10):1117-24

Reviewer 2 Report
Comments and Suggestions for Authors
In general.
The topic is relevant and well-contextualized. Managing the costs of hospital infrastructure is an important challenge for healthcare systems worldwide, and the analysis of the Italian case is interesting.
The objectives are clear and well-defined, although perhaps somewhat broad.
Methodology: The mixed methodology (literature review, financial data analysis, questionnaire) seems appropriate to answer the research questions. However, there are some concerns regarding the statistical robustness and generalizability of the results.
In Details:
Introduction:
The introduction provides good general context but could be more specific in defining "facility management" and its components.
The research questions are clear but perhaps a bit broad. It would be useful to focus on more specific aspects.
Methods:
The description of the sample (27 hospitals in Lombardy) is vague. More details should be provided on the characteristics of the hospitals (size, type, ownership, etc.) and the sampling method.
The sampling method is unclear. Were the hospitals selected randomly? What criteria were used? This affects the generalizability of the results.
The description of the questionnaire is limited. What questions were asked? How was the questionnaire validated?
The statistical analysis is described very generally. It's necessary to specify which statistical tests were used and how statistical significance was assessed, and to know which software was used, and to know if tests of normality were done, and if multivariate models were used.
Results:
The results are presented clearly and concisely, with well-structured tables and figures.
However, the interpretation of the results is sometimes a bit superficial.
The discussion of the differences between the different types of hospitals is interesting but needs further elaboration.
Discussion:
The discussion summarizes the main results well but could be more critical in assessing the limitations of the study.
The practical implications of the results are discussed somewhat generically.
The suggestions for future research are valid but could be more specific.
Tables and Figures:
Figure 6: The y-axis ("Trend of FM Costs") is not clearly defined. What exactly do the values on the y-axis represent? Are they total costs, average costs, or something else? It would be helpful to add a clear unit of measure (e.g., "Millions of Euros").
Table 6: The heading states "Descriptive Statistics of Hospital Facility Management Costs." But what costs are these? Only utilities, only maintenance, or total costs? It is necessary to specify.
The figures are generally clear and legible, but the labels on the axes are sometimes a little small and difficult to read.
Some tables could benefit from additional information, such as the sample size for each type of hospital.
More details should be provided about the statistical analyses used.
Statistical significance tests should be performed to compare costs between the different types of hospitals and to evaluate the trend of costs over time.
Appropriate statistical techniques (e.g., multivariate regression analysis) should be used to control for potential confounding factors.
Potential Confounding Factors: It is unclear whether potential confounding factors (e.g., hospital size, geographic location) have been taken into account. Causality: Causal claims (e.g., relationship between building age and operating costs) must be supported by empirical data.
The limited sample size and unclear sampling method limit the generalizability of the results.
The limitations of the study must be explicitly discussed, including the sample size, the unclear sampling method, and the potential presence of unconsidered confounding factors.
Provide a more precise definition of "facility management" and its components.
Include relevant missing studies in the literature review.
Provide more details about the questionnaire and its validation.
Improve the clarity and legibility of the figures and tables.
English Quality: Carefully review the document to correct grammatical and stylistic errors.
Final Conclusion
The paper has the potential to provide a significant contribution to the understanding of hospital infrastructure costs in Italy. However, the quantitative rigor of the analysis needs to be improved, more details should be provided about the methodology used, and the limitations of the study should be explicitly discussed. The authors are encouraged to carefully consider these suggestions to improve the quality and impact of their work.
Author Response
Thank you very much for the review, the comments ha been important and extremely valuable to improve the manuscript. On behalf of all authors, please find the detailed responses below and the correspinding author revisions.
Comment 1: Facility Management definition Response 1: Facility Management definition A definition has been provided in paragraph 1.3
Comment 2: Introduction The introduction provides good general context but could be more specific in defining "facility management" and its components. The research questions are clear but perhaps a bit broad. It would be useful to focus on more specific aspects. Response 2: Introduction The research questions have been restructured to focus more on the specificity of the study.
Comment 3: Methods Response 3: Methods The method sections has been significantly adapted to clarify the methodology adopted in terms of sampling approach and the questionnaire structures. Also, the statistically analysis approach has been described.
Comment 4: Results The results are presented clearly and concisely, with well-structured tables and figures. However, the interpretation of the results is sometimes a bit superficial. Response 4: Results The different of hospital types has been further elaborated in Paragraph 3.5 |
Comment 6: Discussion:
Response 6: Discussion The discussion section has been revised as requested. Comment 6: Tables and Figures: Table 6: The heading states "Descriptive Statistics of Hospital Facility Management Costs." But what costs are these? Only utilities, only maintenance, or total costs? It is necessary to specify. The figures are generally clear and legible, but the labels on the axes are sometimes a little small and difficult to read. Some tables could benefit from additional information, such as the sample size for each type of hospital.
Response 6 Figure 6 has been revised to clarify the y-axis, which represents the average annual Facility Management (FM) costs per hospital, expressed in millions of euros (€M). The axis label now reads “Average FM Costs (in €M),” and we have added a note to explicitly state that the values refer to the mean total FM costs per hospital per year, including utilities, cleaning, and maintenance services.
Comment 7: Statistical Analysis More details should be provided about the statistical analyses used. Statistical significance tests should be performed to compare costs between the different types of hospitals and to evaluate the trend of costs over time. Appropriate statistical techniques (e.g., multivariate regression analysis) should be used to control for potential confounding factors. Potential Confounding Factors: It is unclear whether potential confounding factors (e.g., hospital size, geographic location) have been taken into account. Causality: Causal claims (e.g., relationship between building age and operating costs) must be supported by empirical data. The limited sample size and unclear sampling method limit the generalizability of the results.
Response 7: Statistical Analysis We appreciate the reviewer’s observation regarding the limited statistical depth of the current analysis. However, it is important to clarify that the collection of primary data regarding hospital infrastructures costs is a very challenging area, where data availability at the level of the single hospital is very limited. Rather, the study is intended to provide a state-of-the-art overview of facility management costs in a sample of Italian hospitals—an area that remains largely unexplored in the existing literature, particularly in the national context. This work represents an initial and innovative attempt to map parametric cost indicators based on real operational data, collected directly from hospitals and healthcare organizations. Given the scarcity of prior benchmarks in this domain, the research prioritizes the establishment of empirical reference values and trends that may serve as a foundation for further investigations. Therefore, the focus has been placed on descriptive and exploratory analysis, which allows for the identification of cost patterns and key areas of variability across hospital facilities of different types and complexity levels. Future research may build upon this work by expanding the sample, applying inferential methods, and integrating multivariate models to explore causal relationships.
|

Reviewer 3 Report
Comments and Suggestions for Authors
The article has a sufficient basic scientific level for publication in a specialized journal, but needs to be improved to increase its scientific value.
There are certain aspects that need to be explained.
The study focuses only on the Lombardy region, which limits the possibility of generalizing to the entire Italian healthcare system, despite the fact that the title suggests a wider national scope. The authors need to explain this point or provide evidence that the data from this region are representative of the entire country.
The article states that extraordinary maintenance costs were excluded from the analysis. It needs to be explained why.
Although there is a general observation that newer facilities have lower costs, Figure 9 shows significant variability that is not fully explained in the discussion.
Some figures are of low quality and informativeness (6, 7, 8, 9)
It is also worth noting that the statistical analysis is not sufficiently deep, statistical significance tests are missing, etc.
There are no comparisons with international indicators to support the conclusions. It would be worth adding a model of factors affecting costs.
Author Response
Comment 1; The article has a sufficient basic scientific level for publication in a specialized journal, but needs to be improved to increase its scientific value.
There are certain aspects that need to be explained.
The study focuses only on the Lombardy region, which limits the possibility of generalizing to the entire Italian healthcare system, despite the fact that the title suggests a wider national scope.
Response 1: The Lombardy region was selected as the focus of this study due to its relevance in the Italian healthcare system: it is the most populous region in Italy, accounting for approximately 16% of the national population, and it hosts the largest number of hospital facilities, including a wide spectrum of typologies (basic hospitals, DEA I, DEA II). Furthermore, the region has a diversified mix of urban and rural healthcare infrastructures, as well as a significant presence of public and accredited private providers, making it a robust case for analysis. While we acknowledge that healthcare systems vary across regions, the diversity and scale of the Lombardy system allow for meaningful insights that may serve as a proxy for national-level trends. We have clarified this rationale in the revised manuscript (see Section 1.6).
Comment 2: The article states that extraordinary maintenance costs were excluded from the analysis. It needs to be explained why.
Response 2: The exclusion of extraordinary maintenance costs from the analysis was a deliberate methodological choice aimed at focusing on the recurring, operational components of facility management expenditures. Extraordinary maintenance typically includes non-recurring capital-intensive interventions, such as major renovations or structural upgrades, which are project-specific and heavily influenced by external funding programs (e.g., PNRR or regional investment plans). Including these would have introduced significant heterogeneity and potential bias in the comparison of annual operating costs across facilities. Instead, by focusing on ordinary maintenance, which is consistent and comparable over time and across institutions, the study ensures a more robust evaluation of the ongoing cost burdens related to hospital infrastructure management
Comment 3: Figures
Although there is a general observation that newer facilities have lower costs, Figure 9 shows significant variability that is not fully explained in the discussion. Some figures are of low quality and informativeness (6, 7, 8, 9)
Response 3: Figures
The text has been updated to clarify the variability in Figure 9. The figures with missing information have been updated. The figure presents the average parametric costs for hospital facility management, disaggregated into three main components: utilities costs, ordinary maintenance, and overall management services. The left panel displays the cost distribution expressed in euros per square meter (€/m²), while the right panel shows the same categories expressed in euros per hospital bed (€/PL). Utilities costs average approximately €129/m² and €18,745/PL, whereas ordinary maintenance shows a lower average of €32.8/m² and €4,610/PL. Management services represent the highest aggregated cost category, with an average of approximately €151/m² and €23,355/PL.
Comment 4: Statistical analysis and international comparison
It is also worth noting that the statistical analysis is not sufficiently deep, statistical significance tests are missing, etc. There are no comparisons with international indicators to support the conclusions. It would be worth adding a model of factors affecting costs
Response 4: We appreciate the reviewer’s observation regarding the limited statistical depth of the current analysis. However, it is important to clarify that the collection of primary data regarding hospital infrastructures costs is a very challenging area, where data availability at the level of the single hospital is very limited.
Rather, the study is intended to provide a state-of-the-art overview of facility management costs in a sample of Italian hospitals—an area that remains largely unexplored in the existing literature, particularly in the national context.
This work represents an initial and innovative attempt to map parametric cost indicators based on real operational data, collected directly from healthcare organizations. Given the scarcity of prior benchmarks in this domain, the research prioritizes the establishment of empirical reference values and trends that may serve as a foundation for further investigations. Therefore, the focus has been placed on descriptive and exploratory analysis, which allows for the identification of cost patterns and key areas of variability across hospital facilities of different types and complexity levels. Future research may build upon this work by expanding the sample, applying inferential methods, and integrating multivariate models to explore causal relationships.

Round 2
Reviewer 2 Report
Comments and Suggestions for Authors
This study addresses an important and significantly under-explored topic within the Italian context, namely the facility management (FM) costs of hospital infrastructures.
The methodology employed, combining financial statement analysis with targeted questionnaires, is appropriate for tackling this issue, even acknowledging certain limitations.
The research successfully provides valuable preliminary quantitative data and benchmarks in a field where such information has been notably scarce for Italy. The manuscript is clearly structured, and the discussion adequately acknowledges the main limitations encountered during the research.
The required revisions are considered minor and primarily focus on enhancing clarity and emphasis in specific areas, rather than necessitating new analyses or fundamental changes to the work. Specifically, the authors are requested to:
-
Place Stronger Emphasis on Sample Size Limitation: While acknowledged, the impact of the low questionnaire response rate (n=27, 20.4%) needs to be more forcefully highlighted in the Discussion and Conclusions. Please elaborate further on how this limits the generalizability of the derived parametric findings (€/sqm, €/bed), even when considering just the Lombardy region.
-
Clarify Sample/Data Source Distinction: Ensure it is explicitly clear to the reader when different data samples are being discussed (e.g., the source for Figure 5 data on building stock age vs. the primary sample of 27 hospitals used for cost analysis).
-
Address In-Text Comments: Please review and address the minor points raised in the specific comments embedded within the PDF manuscript file (e.g., [MR1], [MR2], [MR3]), clarifying the intended meaning or interpretation where necessary.
-
Refine Contextualization: Briefly enhance the discussion around external factors like the COVID-19 pandemic and the energy crisis, acknowledging how these exceptional circumstances might complicate the interpretation of cost trends observed during the 2020-2022 period.
Overall, despite being primarily exploratory and descriptive, this work represents a valuable contribution to the specific Italian context, given the previous lack of detailed data. Addressing these minor points will improve the manuscript's clarity and robustness significantly.
Author Response
Thank you for your valuable suggestions, the issue you raised were very relevant for this final draft of the paper!
Comment 1: Place Stronger Emphasis on Sample Size Limitation: While acknowledged, the impact of the low questionnaire response rate (n=27, 20.4%) needs to be more forcefully highlighted in the Discussion and Conclusions. Please elaborate further on how this limits the generalizability of the derived parametric findings (€/sqm, €/bed), even when considering just the Lombardy regio
Response 1:
lthough the response rate (20.4%, n=27) is aligned with expectations for voluntary technical surveys in healthcare facilities, we recognize that it constrains the generalizability of the derived parametric indicators (€/sqm and €/bed), even within Lombardy. We now state clearly (in line 559 and the following - just before the conclusions of section ) that the values presented should be interpreted as exploratory reference points rather than definitive national benchmarks
Comment 2: Clarify when different datasets are used (e.g., Figure 5 vs. 27 hospital sample).
Response 2: The clarification has been added, please find it from line 384.
Comment 3: Address In-Text Comments: Please review and address the minor points raised in the specific comments embedded within the PDF manuscript file (e.g., [MR1], [MR2], [MR3]), clarifying the intended meaning or interpretation where necessary.
Response 3: In-text comments were internal comments left from different authors, NOT from reviewers. We apoligize for the mistake.
Comment 4: Refine Contextualization: Briefly enhance the discussion around external factors like the COVID-19 pandemic and the energy crisis, acknowledging how these exceptional circumstances might complicate the interpretation of cost trends observed during the 2020-2022 period.
Response 4: the issue regarding the specific historical period with the pandemic has been clarified and expanded to comment on the cost increase in the debated period, as stated from line 441 and the following.
